# Multifaceted Clinical Effects of Echinochrome

**DOI:** 10.3390/md19080412

**Published:** 2021-07-26

**Authors:** Hyoung Kyu Kim, Elena A. Vasileva, Natalia P. Mishchenko, Sergey A. Fedoreyev, Jin Han

**Affiliations:** 1Cardiovascular and Metabolic Disease Center, Smart Marine Therapeutic Center, Department of Physiology, College of Medicine, Inje University, Busan 57392, Korea; estrus@inje.ac.kr; 2Department of Health Sciences and Technology, Graduate School of Inje University, Busan 57392, Korea; 3G.B. Elyakov Pacific Institute of Bioorganic Chemistry, Far-Eastern Branch of the Russian Academy of Science, Prospect 100 let Vladivostoku, 159, 690022 Vladivostok, Russia; vasilieva_el_an@mail.ru (E.A.V.); mischenkonp@mail.ru (N.P.M.); fedoreev-s@mail.ru (S.A.F.)

**Keywords:** histochrome, echinochrome, clinical effect

## Abstract

The marine drug histochrome is a special natural antioxidant. The active substance of the drug is echinochrome A (Ech A, 7-ethyl-2,3,5,6,8-pentahydroxy-1,4-naphthoquinone), the most abundant quinonoid pigment in sea urchins. The medicine is clinically used in cardiology and ophthalmology based on the unique properties of Ech A, which simultaneously block various links of free radical reactions. In the last decade, numerous studies have demonstrated the effectiveness of histochrome in various disease models without adverse effects. Here, we review the data on the various clinical effects and modes of action of Ech A in ophthalmic, cardiovascular, cerebrovascular, inflammatory, metabolic, and malignant diseases.

## 1. Introduction

Polyhydroxynaphthoquinone echinochrome A (Ech A) is a common pigment found in many sea urchin species [1,2]. Its structure, 7-ethyl-2,3,5,6,8-pentahydroxy-1,4-naphthoquinone (molecular weight [MW] = 266), was confirmed via spectral methods [3] and X-ray analysis [4] (Figure 1).

Ech A is a dark red crystalline powder with a melting point of −219 to 221.5 °C. It is moderately soluble in alcohol, very slightly soluble in chloroform, and practically insoluble in water. The ultraviolet–visible (UV–Vis) spectrum of a 0.002% solution of Ech A in ethyl alcohol with hydrochloric acid has a λ_max_ of 342 and 468 nm, λ_min_ of 295 and 392 nm, and two shoulders from 485 to 500 nm and from 515 to 537 nm. The specific absorption coefficient of Ech A $\left(A_{1cm}^{1\%}\right)$ is 272 at 468 nm.

On an industrial scale, Ech A is the active substance (Reg. No. in the Russian Federation P N002362/01) of Histochrome®. It was developed in the G.B. Elyakov Pacific Institute of Bioorganic Chemistry, Far-Eastern Branch of the Russian Academy of Science (Vladivostok, Russia). The active substance was derived from the sea urchin *Scaphechinus mirabilis* [5]. The composition, methods of production, and application of histochrome were patented in Russia, the United States, and the European Union [6,7,8,9].

The initial studies on the antioxidant activity of Ech A and spinochromes A–E were carried out by Boguslavskaya et al. on the model of inhibition of low-temperature initiated oxidation of cumene. Ech A exhibited antioxidant properties with a rate constant of *k*_7_ (4 L/mol*s) when it interacted with the peroxyl radical, whereas the rate constant of Ech A when it interacted with butylhydroxytoluene (BHT) was 2.2 L/mol*s. The results of this study allowed us to present the quinonoid pigments of sea urchins as a new class of natural antioxidants. Furthermore, it has been reported that Ech A exhibits its antioxidant property through various mechanisms, including the scavenging of reactive oxygen species (ROS), interaction with lipid peroxide radicals, chelation of metal ions, inhibition of lipid peroxidation, and regulation of the cell redox potential [10,11,12,13,14,15].

The unique physicochemical properties of Ech A have conferred remarkable therapeutic potential to its clinical form, histochrome. Histochrome is used for the treatment of various diseases, including ophthalmic, cardiovascular, inflammatory, and metabolic diseases. Here, we review the data on the various clinical effects and modes of action of Ech A.

## 2. Clinical Application and Target Molecules of Ech A

### 2.1. Histochrome in the Treatment of Ophthalmic Diseases

Ech A has been shown to successfully resolve traumatic hemophthalmia in rabbits. Additionally, the effect of Ech A on third-degree alkaline eye burns in rabbits has been studied [16]. Rabbits receiving subconjunctival injections of Ech A exhibited a decrease in corneal defects on the 10th day postinjection. In addition, the administration of Ech A improved the epithelialization of the cornea, reduced inflammation of the eyes, and reduced the risk of corneal perforation.

These experimental data served as a prerequisite for the development of the preparation of histochrome. Approximately 0.2 mg/mL (Reg. No. P N002363/02) of the drug was used in the preparations. The drug appeared as a transparent solution with a red–brown color and a pH of 6.5–7.5. The sodium derivative of Ech A and 0.9% sodium chloride solution with no other components was administered for the treatment of degenerative diseases of the retina and cornea, macular degeneration, primary open-angle glaucoma, diabetic retinopathy of the retina, hemorrhages in the vitreous body, retina, anterior chamber, and dyscirculatory disorders in the central and retinal veins. Furthermore, clinical trials approved by the Ministry of Health of the Russian Federation confirmed the efficacy of histochrome in the treatment of hemophthalmos of various origins, corneal chemical burns, diabetic retinopathies, dystrophic lesions of the retina, uveitis, keratitis, and cataracts. The results of histochrome applications in ophthalmology have been described in the various studies [17,18,19].

Many pathological processes in the eyes occur due to the increase in the number of free radicals and the appearance of lipid peroxidation products [20]. Antioxidants such as tocopherol acetate, vitamins C and A, and dicynone are widely used as inhibitors of free radical processes in ophthalmology. However, their effectiveness in treating a number of medical conditions is inadequate.

Clinical trials that explored the efficacy of histochrome revealed that the clinical effect of the drug was based on the localization, prescription, and the extent of hemorrhage [21]. A total of 92 patients aged 16–85 years with various diseases, including hemophthalmos, thrombosis of the central retinal vein and its branches, preretinal hemorrhages, central degeneration of the retina, hyphema, and dystrophia of the cornea, were studied, and an increase in visual acuity after resorption of small hemorrhages in the anterior parts of the vitreous humor was observed in 73.6% of the cases. Furthermore, these cases showed a tendency to decrease exudation, macular edema, and hemorrhage resorption on the fundus. In addition, studies on the peripheral and central fields of vision revealed a decrease in the number of scotomas, while the remaining scotomas were mostly relative.

When histochrome was applied for the treatment of hyphemia, almost all patients showed positive dynamics, which were characterized by a rapid reduction in the imbibition of the iris and the posterior surface of the cornea. There was a complete resolution of the total hyphema after two injections in 15.3% of the cases and after three to five injections in 77% of the cases. According to the data obtained via electroretinography, histochrome has pronounced retinoprotective properties.

There was an increase in visual acuity in the majority of patients with central degeneration of the retina after treatment with histochrome. Moreover, the condition of all patients with dystrophic diseases of the cornea improved. In particular, in these patients, a reduction in pain syndrome was observed. Decreased edema, epithelialization of defects, and increased visual acuity were objectively observed in 60% of the cases.

Thus, the authors of this large-scale clinical study conclude that the drug was effective in 92.8% of the cases with proliferative processes, degeneration, and hemophthalmos. Histochrome facilitates hemorrhage resolution, protects the retina, and results in antioxidation. These properties confer the drug the ability to effectively treat diseases caused by metabolic disorders in the retina, uvea, and cornea, to improve tissue tropism, to eliminate edema, and to accelerate epithelialization [21].

The effectiveness of histochrome was explored in a study involving 213 patients with various eye pathologies, including hemophthalmia and retinal hemorrhages of various etiologies (*n* = 44), postoperative hyphema (*n* = 12), diabetic retinopathy (*n* = 96), keratitis (*n* = 28), endothelial-epithelial dystrophia of the cornea (*n* = 3), postoperative corneal edema (*n* = 15 people), central retinal vein thrombosis (*n* = 5), and retinal dystrophy (*n* = 10) [22]. Histochrome (0.5 mL) was used in the form of a parabulbar or subconjunctival injection. Moreover, the treatment course consisted of around five to ten injections. In addition, in patients with keratitis and corneal dystrophies, histochrome was used in the form of drops consisting of 0.02% solution. Approximately one drop was administered four to six times per day. The most significant therapeutic effect of histochrome was observed in postoperative hyphema, which completely resolved after three to four injections. Histochrome demonstrated a therapeutic effect when it was applied for the treatment of hemophthalmia and retinal hemorrhages of various origins (post-traumatic, postoperative, postinflammatory, myopia, retinal detachment, or hypertonic angioretinopathy). The therapeutic efficacy was especially apparent when treating post-traumatic and postoperative hemorrhages. Recent hemophthalmos not accompanied by serious eye pathology were associated with a more rapid and complete resolution. However, in all cases of hemophthalmia, positive dynamics were observed. The most numerous group was that consisting of patients with diabetic retinopathy (*n* = 96). The group included patients with nonproliferative, preproliferative, and proliferative diabetic retinopathy. Histochrome was administered to all of them as a subconjunctival or a parabulbar injection, with at least 10 injections. Furthermore, it was found that in patients with expressed retinal lipoidosis, there was a decrease in solid exudates and a disappearance of the soft exudates. When treating hemorrhages in the retina and vitreous body, blood was mostly resorbed in the anterior vitreous layer and slightly in the posterior vitreous layer and retina. The effect of diabetic retinopathy on hemorrhage may be associated with the parabulbar method of histochrome injection, especially if the injection reaches only the anterior segment of the eye. In addition to resolving the effects of histochrome when treating DR, improvements in some biochemical blood parameters in patients with diabetes mellitus were observed (normalization of the total cholesterol (TC), triglyceride (TG), and low-density lipoprotein cholesterol (LDL-C) levels). This fact explains the improvement in the condition of the fundus in patients with DR, when exudates disappear and hemorrhages on the retina diminish. Positive results were also obtained when keratitis and endothelial-epithelial dystrophy (EED) of the cornea were treated with histochrome. In these cases, histochrome was administered to patients as a combination of droplets and subconjunctival injections. A clinical effect was observed when histochrome was administered to patients with central retinal vein thrombosis.

The authors determined the groups of eye diseases that can be treated with the antioxidant drug histochrome, namely hyphema and hemophthalmia of various etiologies, including diabetic retinopathy, inflammatory keratitis, injuries and operations, epithelial-endothelial dystrophy of the cornea, and retinal dystrophy, especially those caused by atherosclerosis with disordered lipid metabolism. Histochrome can be used locally as a monodrug in many cases. The results show that histochrome facilitates hemorrhage resolution and improves metabolic processes in the retina, uvea, and cornea, and has a lipotropic effect. The drug has no side effects and can be recommended for application in ophthalmology.

In patients with central retinal vein (CRV), massive hemorrhage occurs in the retina containing a large amount of free iron. According to a study, histochrome, which possesses antioxidant and chelating properties, is an effective modern drug [23]. Accumulated clinical experience and obtained data on the increase in the processes of lipid peroxidation in patients with occlusion of CRV and its branches demonstrated that the efficacy of histochrome in retinal venous occlusion was associated with changes in the redox system.

Histochrome was applied for the complex treatment of 38 patients aged 52–76 years with initial and advanced stages of central chorioretinal dystrophy (CCD) [24]. Approximately 5 mL of 0.02% solution of the drug was injected into the posterior pole of the eye with preliminary trophic sclerectomy in the lower outer segment of the eyeball, and the treatment course lasted for 10 days. The observed increase in the parameters of local electroretinography in 28 patients indicated a significant improvement in the metabolic processes in the retina and pigment epithelium. Moreover, the authors observed positive dynamics in the fundus that were characterized by the resorption of small hemorrhages, increased arterial diameter, and a decreased number of soft drusen. In 86% of patients with the advanced stage of CCD, application of histochrome resulted in a significant improvement in visual function. This might be attributed to the improvement in blood flow in the posterior eye, which was observed in a large number of patients.

The effectiveness of surgical treatment with the simultaneous application of histochrome and magnetotherapy during sinus trabeculectomy (STE) and the early postoperative period in patients with primary open-angle glaucoma (POAG) has been studied. This complex treatment led to the improvement of visual function and stabilization of the delayed glaucoma process within a six-month observation period [25] (Figure 2).

### 2.2. Histochrome in Pediatric Practice

#### 2.2.1. Application of Histochrome in Pediatric Ophthalmology

The effectiveness and safety of histochrome in pediatric patients was studied in a group of 554 children, aged 1 to 14 years, with second and third degree traumatic hyphema, hemophthalmos, hemorrhages in the fundus, hemorrhagic retinovasculitis, diabetic retinopathy, uveitis with a pronounced exudative component, and edema of the retina with hemorrhages in the anterior chamber after reconstructive operations with cataract extraction, hemorrhages, and exudative changes in the back segment of the eye in patients with retinopathies of prematurity [26,27]. Histochrome, especially when administered through irrigation systems, was found to be an effective treatment for various degrees of hemophthalmia. In these cases, the treatment period was short, which resulted in the retention of retinal function. However, histochrome treatment did not result in gross vitreous destruction. A less pronounced effect was observed in patients with retinal pathology accompanied by general and local pathology, owing to the duration of the process and destructive changes in the retina. However, even when such gross changes took place in the eyeball, the active resolution of hemorrhages commenced on the third day of histochrome treatment. In particular, good results were obtained when recent hemorrhages in the retina were treated. Based on the results obtained, the following conclusions were drawn: (1) 0.02% of histochrome solution facilitates active blood resorption in the eyeball in patients with traumas and in children of different ages with diseases of the inner coats of the eye with hemorrhagic components; (2) the effectiveness of histochrome treatment depends on the stage of the disease and is potentiated by administration at an earlier stage and injection into the retrobulbar space through the irrigation system; (3) histochromes can be used to treat retinopathy with retinal edema; (4) the administration of histochrome for the complex treatment of hemophthalmia and hemorrhage into the retina can improve its functionality owing to its retinoprotective effect; (5) drug application does not result in any adverse effects in the patient; (6) histochrome is clinically recommended for the treatment of intraocular diseases of various origins, retinopathy with a pronounced exudative component, and uveitis (Figure 2).

#### 2.2.2. Use of Histochrome in Premature Infants

Premature infants are at an increased risk of diseases caused by active ROS, including retinopathy of prematurity (RP), bronchopulmonary dysplasia, necrotizing enterocolitis, and periventricular leukomalacia. This is primarily due to the immaturity of the antioxidant system and the low efficiency of homeostatic mechanisms that protect cells from the damaging effects of oxidative stress [28].

A study exploring the effect of intramuscular administration of 0.02% histochrome solution to children with pre-plus disease and plus disease of active RP was conducted. The children in the primary and control groups were similar in terms of somatic parameters and the severity of RP. Histochrome in the form of intramuscular injections in children with extensive avascular zones in preretinopathy and prethreshold stages of plus disease and pre-plus disease resulted in the following: (1) increase in the incidence of spontaneous regression of the disease (main group: 44.0%, control: 12.5%); (2) delayed progression of the process to the threshold stage, which was defined as the period from the detection of RP to the advancement of the disease and was 0.7–9 weeks (4.11 ± 0.57) in the main group and 0.6–3.5 weeks (2.28 ± 0.19) in the control group, thereby causing reduction in the area of the avascular zones in the retina by the time of laser coagulation; accordingly, the severity of cicatricial changes in the future enabled operation of an older child with less somatic complication; and (3) increased efficiency of laser coagulation at the threshold stage of the disease [29].

In a study including premature infants born at the 27th (27.0 ± 2.5) gestational week, weighing 914 ± 247 g at the time of birth, 282 children (564 eyes) were examined. Prophylactic use of antioxidants such as histochrome in these children who were at risk of RP showed that the occurrence of RP was significantly reduced compared with those who were not or those who received the antioxidant drug emoxipine instead. Histochrome was prescribed for RP therapy in the form of parabulbar injections; therefore, a forced instillation of a solution for injections was made. The study showed that forced instillation of 0.02% of histochrome solution significantly reduced the occurrence of RP, especially stage III, and thus contributed to more favorable outcomes. The authors concluded that histochrome (pentahydroxyethylnaphthoquinone) was an effective and safe drug for the treatment of nonproliferative stages of retinopathy of prematurity and vitreous hemorrhage [30] (Figure 2).

#### 2.2.3. Application of Histochrome for the Treatment of Adolescent Children

To obtain permission for clinical trials of histochrome in pediatric practice, an experimental study of the drug in animals at an appropriate age was required.

A large-scale study on the effect of Ech A on the antioxidant status of various systems was examined via the chemiluminescence method. The following parameters of the experimental and clinical biomaterials were determined: S_sp_, the intensity of free radical oxidation, including superoxide anion (S_luc_) and hydroxyl radical (S_lum_); h, the content of lipid hydroperoxides; Sind-1, the rate of accumulation of peroxide radicals; H, the inverse of peroxide resistance; and Sind-2, the inverse of antioxidant protection activity.

According to the results of the chemiluminescence analysis of the free radical status dynamics of the blood serum, Ech A treatment (0.5 mL of 0.02% histochrome solution injected intramuscularly for five days) in children with chronic infiltrative lung disease (CILD) resulted in a 1.6-and 1.9-fold decrease in Sind-2 and H levels, respectively, which was due to the increased activity of antioxidant protection [31]. Thus, in children with CILD, Ech A treatment corrected the disorders of systemic free radical status. An increase in the relative CD3+ (T-lymphocyte) and CD4+ (T-helper-inducer) cell numbers, a decrease in CD8+ (T-cytotoxic lymphocyte) cell number, and an increase in the immunoregulatory index were observed in children with CILD four to five weeks after treatment with Ech A [32].

The same group of scientists showed the effectiveness of Ech A application in the treatment of erosive gastroduodenitis in adolescents [33]. In the homogenized bioptates of the gastroduodenal mucosa of adolescents who received intramuscular injections of Ech A (2 mL of 0.02% histochrome solution for two days), the accumulation of peroxide radicals (Sind-1) and lipid hydroperoxide (h) decreased by 1.4- and 1.5-fold, respectively, compared with the control group of patients. The reduction in free radical formation (Ssp) in patients who received Ech A injection was 2.2 times more intensive in the general population than that in adolescents undergoing standard therapy. On the 10th day after treatment, gastric erosion was healed in 54% of patients in the control group and in 85% of patients treated with Ech A (Figure 2).

### 2.3. Histochrome in Cardiovascular Disease

#### 2.3.1. Ischemic Heart Disease

Ischemia occurs when blood supply to the heart is reduced due to blood loss, blockage or damage of the coronary artery, and recovery of blood flow after ischemia, which is defined as reperfusion. Reperfusion within 20 min after the onset of ischemia is reversibly inhibited by the internal homeostasis of the heart, but reperfusion after ischemia for more than 30 min causes irreversible ischemia and reperfusion damage of the heart. This damage begins with direct necrosis and apoptosis of cells that leads to structural remodeling of the heart, which includes fibrosis and hypertrophy of the heart. These processes eventually culminate to death due to heart failure. Thus, inhibition of the exponential increase in ROS levels during ischemia and reperfusion and prevention of fibrosis due to the inflammatory response of the heart are important targets for the treatment of ischemic heart disease. Therefore, the strong antioxidant and anti-inflammatory effects of Ech A have high potential as a treatment for ischemic heart disease; thus, various studies have been conducted to determine the effectiveness of Ech A treatment for ischemic heart disease (Figure 2).

The cardioprotective properties of Ech A were first discovered in occlusive reperfusion myocardial infarction in acute open-chest experiments on dogs [34]. It was shown that Ech A, when administered via the intravenous route (1 mg/kg), restored the strength of contractions and tension at rest. In the initial period of reperfusion, Ech A reduced the release of creatine phosphokinase, a cardiac isoenzyme, into the coronary effluent, the level of which increased sharply when myocardial cells were damaged. Moreover, Ech A treatment significantly reduced the size of the area of the myocardial infarction. In a model of myocardial infarction in dogs caused by ligation of the descending branch of the left coronary artery, a single intravenous administration of Ech A at a dose of 1 mg/kg reduced the size of the necrotic zone by 40% compared with that of the control group [34].

These experimental data served as a prerequisite for the development of the preparation histochrome for intravenous administration. A 10 mg/mL (Reg. No. P N002363/01) dark brown solution with a pH of 7.0–8.0 containing a sodium derivative of Ech A without additives was obtained under inert conditions for the treatment of coronary heart disease to limit the zone of necrosis in myocardial infarction. Multiple clinical studies reported the beneficial effect of histochrome in patients with ischemic heart disease in Russia [35,36,37,38,39,40,41].

Zakirova et al. tested the clinical efficacy of histochrome in 66 patients with unstable angina pectoris (UAP). Histochrome treatment reduced lipid peroxidation and aggregation of platelets in the blood of UAP patients. These results provided evidence of the strong antioxidant effect and clinical efficiency of histochrome in patients with heart diseases [35]. In a subsequent study, the authors evaluated the efficacy of histochrome in suppressing myocardial infarction and in modulating the activity of cardiac creatine phosphokinase in 45 patients with acute myocardial infarction treated with thrombolytic therapy [39]. This study also evaluated the pharmacokinetics of histochrome in seven patients with myocardial infarction after a single intravenous infusion of 100 mg of the drug. Treatment with histochrome reduced the area of myocardial infarction and enhanced the myocardial contractility of patients [39].

ATP is an essential bio-energy molecule in cells, and the balance of the intracellular ATP level is important for cardiac function. Afanas’ev et al. tested the effect of histochrome on ischemia-induced depletion of ATP in the myocardium of patients with heart disease. Cardiac tissues were obtained from patients with coronary heart disease (CHD; *n* = 17) or non-CHD (*n* = 6; controls). Of the 17 CHD patients, 8 received two intravenous injections of 3% of histochrome at a dose of 1 mg/kg 24 h prior to the operation [36]. The surgically obtained tissues were exposed to ischemia for 60 min, and changes in ATP levels were observed. Tissues from the nontreated CHD patients were extremely vulnerable to ischemia treatment; their ATP level fell to 12% of the basal level. Contrastingly, tissues of patients receiving histochrome had maintained ATP at 80% of the basal level [36].

In a subsequent clinical study, Afanas’ev et al. reported that histochrome reduced oxidative stress in the heart and blood vessels of patients with ischemic heart disease [38,40]. Treatment with histochrome reduced lipid peroxidation, a molecular marker of oxidative stress, and levels of MB-creatine phosphokinase, which was released in the blood of patients with chronic ischemic heart disease during and after operation for aorto-coronary shunting [38]. In addition, histochrome treatment lowered the level of blood lipid peroxidation, which was measured to be high in patients with severe cardiac failure (Functional Class IV) compared with those with mild functional class II [40].

To investigate the molecular mechanism of the cardioprotective effect of Ech A in ischemic heart disease, studies using animal models of heart disease were conducted [42,43,44,45]. In addition to its antioxidant effect, these studies investigated the role of Ech A in intra-myocardial Ca^2+^ regulation and electrophysiological alterations during periods of ischemia and reperfusion. During ischemia and reperfusion, cardiomyocytes undergo a rapid change in intracellular Ca^2+^ concentration due to the loss of Ca^2+^ pump function and ATP. Accordingly, Ca^2+^ overload in cardiomyocytes causes an electrophysiological imbalance of the heart and impairs mitochondrial function, thus resulting in heart arrhythmia and impaired contractile/relaxation capacity [42,46]. Ex vivo ischemia mimetic Ca^2+^ depletion and reloading treatment resulted in the loss of ATP, phosphocreatine, myoglobin, and the uncoupling of mitochondria in isolated rat hearts. Pretreatment with histochrome attenuated cardiac and mitochondrial abnormalities caused by Ca^2+^ dysregulation [42]. In a study of arrhythmia induced by ischemia in cats, treatment with Ech A suppressed the incidence of arrhythmia and the change in repolarization during arrhythmia [44,45]. Moreover, a new mechanism for the regulation of cardiac Ca^2+^ signaling by Ech A was suggested in our previous study [43]. In ex vivo studies of isolated rat hearts and cardiomyocytes, Ech A mildly inhibited the Ca^2+^ uptake of SERCA2A through the regulation of phosphorylation of phospholamban. Inhibition of SERCA2A by Ech A was shown to suppress cardiac toxicity due to calcium overload [43]. The inhibitory effect of Ca^2+^ ATPase in the sarcoplasmic reticulum by Ech A probably supports the mechanism of the previously reported ATP-sparing effect of Ech A under conditions of ischemia and reperfusion [36].

#### 2.3.2. Ischemic or Hemorrhagic Stroke

A stroke is a state of complete loss or significant reduction in blood supply to the brain, which causes irreversible brain damage. There are two types of stroke: ischemic and hemorrhagic. Ischemic stroke occurs when blood flow to the brain is decreased due to the narrowing of blood vessels, such as arteriosclerosis. In contrast, hemorrhagic stroke is caused by the rupture of a blood vessel, resulting in a decrease in blood volume to the brain and an increase in blood pressure in the surrounding area, which is referred to as cerebral edema [47].

Studies have evaluated the neuroprotective effect of Ech A in animal models of hemorrhagic or ischemic stroke [48,49,50,51]. Histochrome (0.7 mg/kg, bolus intraperitoneal injection) effectively reduced the edema caused by hemorrhage, which thus alleviated neurological symptoms in rats [48,51]. In the histochrome-treated group, the edema decreased more rapidly within 48 h after hemorrhagic treatment compared to that observed in the control group. In addition, in the histochrome-treated group, edema almost completely disappeared within seven days after the operation. The authors suggested that the strong anti-inflammatory effect of Ech A facilitated the reduction in edema and subsequent brain damage. Furthermore, we confirmed that intraperitoneally administered histochrome was detected in the cerebrospinal fluid (CSF) of the brain, confirming the positive therapeutic effect of the systemically administered drug.

Additionally, Ech A increased the blood flow in the cerebral blood vessels damaged by long-term ischemia in the senescence-accelerated OXYS rat model, resulting in the improvement of brain function and anxiety [50]. Briefly, chronic ischemia reduced the diameter of the coronary artery (CA) in senescence-accelerated OXYS rats. Histochrome (1% Ech A solution) was injected intraperitoneally (1 mg/kg) for five days. The same doses of additional injections were administered after two months. Interestingly, Ech A treatment specifically enhanced the CA diameter and blood flow in OXYS rats compared to control Wistar rats, suggesting that Ech A was a general vasodilator that improved disease-specific vascular function [50].

The most recent study showed the therapeutic efficiency of Ech A in ischemic stroke and hemorrhagic stroke [49]. The brain ischemia and reperfusion model was induced by 90 min of middle cerebral artery occlusion (MCAo). Ech A (10 μM) was infused through the cerebral artery at the withdrawal point in the MCAo rat models. Treatment with Ech A reduced the infarcted brain area and immobility time in the forced swim test. Ech A reduced the apoptotic signaling, including Bax and caspase-3, and enhanced survival signaling, such as the ERK/Akt pathway [49] (Figure 2).

#### 2.3.3. Cardioprotection Against Cardiotoxins

Several therapeutic drugs, such as sodium nitroprusside (SNP), a hypertensive drug, and doxorubicin, an anticancer drug, can induce oxidative stress and cardiotoxicity, which are adverse risk factors for heart failure [52,53]. Thus, we had assessed the cardioprotective effect of Ech A and compared it with SNP and doxorubicin in metabolic stress-induced rats and human cardiac cell lines [54,55].

Cardiotoxins significantly increased mitochondrial dysfunction, ROS levels, and membrane potential depolarization and decreased the ATP production in H9C2 cells. Mitochondrial damage and cell death were attenuated by Ech A treatment (1 or 3 μM). The cardioprotective effect of Ech A was facilitated by the downregulation of ERK/JNK and p38 cell death signaling pathways [55]. Ech A showed strong antioxidant and cardioprotective effects against various types of stress, including oxidative stress (1 mM of H_2_O_2_ or 0.1 μM of doxorubicin), metabolic stress (10 mM d-galactose or 33.3 mM of glucose), and hypoxic stress (10 μM of CoCl_2_) [55].

To understand the mitochondrial and cardiac protective mechanisms of Ech A, we investigated the effect of Ech A on mitochondrial biology, including oxidative phosphorylation, ATP production, and mitochondrial biogenesis and found that 2.5–100 μM Ech A did not affect viability or the mitochondrial inner membrane potential of H9C2 cells treated with Ech A [55]. Interestingly, Ech A (10 μM for 24 h) enhanced mitochondrial oxidative phosphorylation capacity, due to the increased mitochondrial mass of cells, and reduced the production of ROS. The 10 μM dose of Ech A increased the mitochondrial mass by 150%. Furthermore, Ech A treatment increased the mRNA and protein levels of mitochondrial biogenesis regulators, such as NRF-1 and PGC-1α, thereby suggesting a potential transcriptional boosting effect of Ech A in mitochondrial biogenesis [56] (Figure 2).

### 2.4. Ech A in Inflammatory and Fibrotic Disease

Owing to its remarkable antioxidant and anti-inflammatory effects, Ech A has been applied to various inflammatory and fibrotic diseases [57,58,59,60,61,62]. A study conducted in Japan demonstrated the anti-inflammatory properties of Ech A in a model of lipopolysaccharide-induced uveitis in rats [57]. The level of TNF-α, a proinflammatory cytokine, in the intraocular fluid of rats with experimental uveitis was significantly decreased when Ech A was injected intravenously. Doses of 0.1, 1, and 10 mg/kg resulted in the reduction of TNF-α levels to 405.4 ± 26.0, 276.0 ± 12.0, and 225.7 ± 9.4 pg/mL, respectively. However, TNF-α levels remained high, at 457.0 ± 29.0 pg/mL, in rats who did not receive any treatment. Moreover, the nuclear translocation of NF-κB p65 in the irido-ciliary zone was significantly suppressed by one injection of Ech A at a dose of 10 mg/kg. According to the results of immunohistochemical studies, the NF-κB inhibition by Ech A was due to a significant decrease in the production of ROS in the eye tissue [57].

The anti-inflammatory and anti-infectious effects of Ech A in various organs have been investigated [58,61,62]. In the gastric ulcer model, Ech A showed a significant antiulcer effect, which was proven by the reduced area of ulceration when the gastric juice volume and acidity were decreased [62]. In addition, Ech A suppressed the infiltration of inflammatory cells such as neutrophils and gastric hemorrhage in the gastric mucosa. As the molecular mechanism of this protection, Ech A treatment significantly reduced oxidative stress (methylenedioxyamphetamine) and increased antioxidant (glutathione [GSH] and superoxide dismutase [SOD]) activities.

Ech A exhibited hepatoprotective effects in animal models of sepsis and intrahepatic cholestasis [58,61]. In a cecal ligation and puncture (CLP) sepsis rat model, Ech A administration significantly protected liver function against sepsis injury. The activities of liver stress marker enzymes, including alanine transaminase (ALT), gamma-glutamyl transferase (GGT), lactate dehydrogenase (LDH), aspartate transaminase (AST), and alkaline phosphatase (ALP), were significantly suppressed by Ech A treatment. Ech A attenuated the hepatic oxidative stress induced by CLP by markedly increasing the GSH content and antioxidant enzyme activities of SOD, catalase, glutathione peroxidase, glutathione reductase, and glutathione-S-transferase). These antioxidant effects of Ech A successfully reduced lipid peroxidation and tissue damage in liver tissues of the CLP model [61].

Ech A showed similar hepatoprotective effects in the rat model of intrahepatic cholestasis [58]. Cholestasis was induced via a single injection of alpha-naphthylisothiocyanate (ANIT) at a dose of 75 mg/kg of body weight. In this study, Ech A (1, 5, and 10 mg/kg body weight) was orally administered to the rats 48 h before the ANIT injection. The hepatoprotective effect of Ech A was also compared with that of oral supplementation of ursodeoxycholic acid (UDCA) at 80 mg/kg of body weight. Ech A significantly decreased the activities of serum AST, ALT, and ALP and the levels of total protein, total bilirubin, direct and indirect bilirubin, and liver MDA and nitric oxide [58].

Prolonged hyperlipidemia and hyperglycemia, obesity, and type 2 diabetes mellitus are strong inducers of inflammatory diseases. Studies have also assessed the beneficial effects of Ech A in various metabolic and inflammatory diseases [59,63,64,65,66,67,68]. The first study to evaluate the beneficial effects of Ech A in a hyperglycemic and hyperlipidemic model was conducted in 2013 [66], wherein a model of diabetes induced by alloxan administration was produced, and treatment with Ech A resulted in the reduction of blood glucose levels and metabolic parameters such as MDA and TG in the diabetic animal model. In addition, levels of liver damage indicators, such as ALT and AST, were significantly reduced in the Ech A-treated group.

In a series of studies, Soliman et al. showed the antidiabetic effect of Ech A in type 1 and type 2 diabetes animal models [67,68]. Ech A-treated streptozotocin (STZ)-induced diabetic rats showed a significant decrease in the body weight with better neuromuscular responses against heat stress and wire suspension. In addition, treatment with Ech A enhanced the protein and lipid profiles of both diabetic models. Moreover, total protein and albumin levels were significantly increased in the Ech A-treated group, while TG, TC, and LDL-C levels were significantly decreased [68].

In a similar setting, administration of Ech A also improved liver function, kidney function, lipid profile, and antioxidative effect in both type 1 and type 2 diabetes mellitus. The antidiabetic efficacy of Ech A was superior in type 1 compared to that in type 2 diabetes [67]. Furthermore, the lipid-lowering effect of Ech A was tested in high fat diet (HFD)-induced hyperlipidemic rats and was compared to that of atorvastatin (ATOR), a known hypolipidemic drug. In this study, Ech A showed similar or slightly superior lipid-lowering and protective effects on liver and kidney functions compared to those of ATOR. The group co-treated with Ech A and ATOR did not show a superior protective effect [63].

In a recent review, the authors suggested that Ech A might exert its antidiabetic effect through four major processes: improved glucose homeostasis, increased insulin production, regeneration of pancreas, and decreased insulin resistance [64]. Thus, we hereby describe the detailed mode of action of the antidiabetic effect of Ech A.

The influence of Ech A on *p53* gene expression was studied using a model of immobilization stress [69]. The expression of *p53* in the red bone marrow of mice exposed to stress was seven times higher than that of intact mice. *p53* expression was 2.5-fold higher in animals that were treated with Ech A at a dose of 1 mg/kg for a week compared with that of the controls. The data obtained demonstrated the antistress properties of Ech A and its ability to influence the expression of the universal cell cycle regulator p53 [69].

Recent studies have discovered novel mechanisms and molecular targets of the anti-inflammatory effects of Ech A [59,60,70]. Ech A effectively inhibited inflammatory bowel disease in mice with colitis. Moreover, Ech A induced the generation of T-regulatory cells and M2 macrophages (anti-inflammatory regulatory cells) through the inhibition of TNF-α, an inflammatory factor, and the activation of IL-10, an anti-inflammatory factor [60]. This M1/M2 macrophage modulation-mediated anti-inflammatory effect of Ech A was also found in bleomycin-induced scleroderma, an inflammatory skin disease, in an animal model [70]. Ech A treatment effectively inhibited scleroderma-induced skin thickness and collagen deposition and inhibited the proliferation of inflammatory M1 and M2 macrophages. TNF-α and IFN-gamma levels, which are significantly increased in scleroderma, were remarkably suppressed by Ech A [70].

In 2020, a randomized clinical trial was conducted in Russia to investigate the anti-atherosclerotic effect of Ech A in 140 patients with cardiovascular diseases [59]. Ech A significantly improved lipid metabolism, enhanced the antioxidative effect, and reduced atherosclerotic inflammation in all treated patients. Treatment with Ech A improved the function of the intracellular matrix and decreased epithelial dysfunction in patients. This study revealed that Ech A increased the surface expression of human leukocyte antigen DR isotype (HLA-DR) and regulated the function of aryl hydrocarbon receptor (AhR) through direct binding. DR, HLA-DR, and AhR are major regulators of immune and inflammatory responses in cardiovascular diseases [59]. These experimental and clinical studies suggest that Ech A is an efficacious drug for the treatment of various inflammatory/immune diseases (Figure 2).

### 2.5. Ech A in Stem Cell Therapy

ROS and inflammatory signals are important factors in regulating stem cell differentiation and growth [71]. Thus, we investigated the effect of Ech A on the differentiation and functional regulation of stem cells [72,73,74].

First, we studied the efficiency of differentiating cardiomyocytes from mouse embryonic stem cells using Ech A. Ech A improved the cardiomyocyte differentiation rate and the beating ability of myocyte-like cells in a dose-dependent manner. Ech A at a concentration of 50 μM improved mitochondrial function in differentiated cardiomyocytes. The observed improvement was independent of mitochondrial ROS, and the selective binding of Ech A with PKCi and subsequent functional inhibition of Ech A were confirmed as the mechanism for enhancing cardiomyocyte differentiation by Ech A [73]. This study identified the atypical PKCi as a novel molecular target of Ech A in stem cell engineering and therapy.

Cardiac progenitor cells (CPCs) are stem cells that exist in small numbers in the heart and play an important role in repairing heart damage. The protective effect of Ech A on human CPCs under H_2_O_2_-induced oxidative stress has been studied [72]. In this study, Ech A suppressed the death of CPCs via oxidative stress by reducing H_2_O_2_-induced oxidative stress in both cells and mitochondria without affecting the differentiating potential of CPCs. Moreover, Ech A effectively inhibited the senescence of CPCs induced by prolonged replication. These results suggest that Ech A is a candidate drug that can increase the survival rate after transplantation of CPCs and increase the efficiency of myocardial cell regeneration [72].

Hematopoietic stem and progenitor cells (HSPCs) are important sources of regenerated blood cells, but their proliferation in an in vitro environment performed for therapeutic purposes is largely limited by extracellular oxidative stress. Ech A, a safe drug, has already been approved for clinical practice and has great advantages in ex vivo expansion and subsequent transplantation of HSPCs into patients. Since Ech A exhibited strong antioxidant and anti-inflammatory effects, the mechanism related to the enhancing effect of Ech A on the ex vivo expansion of HSPCs was studied [74]. Ech A suppressed ROS generation and p38-MAPK/JNK phosphorylation, which enhanced the expansion of CD34+ HSPCs. Ech A induced the ex vivo expansion of CD34+ cells via Src/Lyn-mediated p110δ expression, suppression of ROS generation, and p38-MAPK/JNK activation [74]. Ech A treatment maintained the stem cell potential of HSPCs during ex vivo expansion.

Ech A effectively protected the survival rate and preserved the stem cell potential of various stem cells in the in vitro and ex vivo expansion processes essential for stem cell treatment. In addition, it is a new safe cell therapy that can facilitate myocardial cell regeneration during transplantation of stem cells or cardiac progenitor cells by promoting differentiation into cardiomyocytes (Figure 2).

### 2.6. Ech A in Cancer Therapy

Although few studies have been performed on the anticancer effect of Ech A, they suggest new clinical applications for this drug in cancer therapy [75,76]. The anticancer effect of Ech A in Ehrlich ascites carcinoma (EAC) was tested in an Egyptian group in 2021 [75]. In addition, Ech A (1 mg/kg body weight, intraperitoneal injection) reduced the tumor volume, improved hematological parameters, and preserved liver and kidney functions in an EAC cancer mouse model [75].

Although the exact mechanism of the anticancer effect of Ech A remains unclear, the inhibitory effect of PKCi [73,77] and the regulatory and binding effects of HLA-DR and AhR [59,78,79,80] are expected to be target candidates for the anticancer effect of Ech A.

In addition to its direct anticancer action, Ech A plays a role in the suppression of the cardiac and mitochondrial toxicity of doxorubicin, a widely used anticancer drug. Hence, Ech A can serve as an adjuvant treatment for chemotherapy [54,55,76] (Figure 2).

## 3. Perspective and Conclusions

Even though Ech A looks like a simple molecule it is not so simple. First of all, it is highly oxygenated, oxygen makes up 42% of the mass of the molecule in contrast to most other secondary metabolites, especially of terrestrial origin. Second, Ech A has the ethyl side chain that makes the molecule lipophilic and able to enter membranous structures. Due to these structural features Ech A also become ionizable under physiological conditions and forms eight possible protomeric structures [81]. Most likely that such a variety of strong effects is explained exactly by the unicity of the structure of Ech A.

In the past few decades, histochrome, a drug formulation of Ech A, has shown limited clinical use in Russia despite its various pharmacological effects. In most of the studies presented in this review, histochrome treatment was compared with standard therapy and with other antioxidant drugs (in some studies) and has proven its advantage. The study results on the various pharmacological actions of Ech A and its derivatives provide important theoretical evidence for expanding the clinical application of histochrome. To expand the clinical application of Ech A, in-depth and robust studies on various new molecular targets, along with its antioxidant and anti-inflammatory properties are required.

The development of new drug forms of Ech A should consider various components that can be added to increase its solubility in water and to provide targeted and controlled release of the drug while preserving or enhancing its pharmacological properties, to expand the application of drugs of interest. Although Ech A was discovered in 1927, it is still waiting for a young challenge and is playing a role as a flagship of new drug development based on natural marine products.

## Figures and Tables

**Figure 1 marinedrugs-19-00412-f001:**
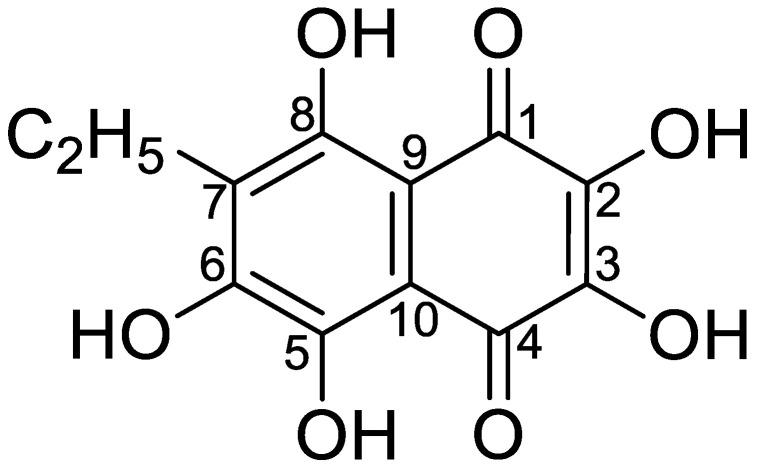
The chemical structure of Ech A.

**Figure 2 marinedrugs-19-00412-f002:**
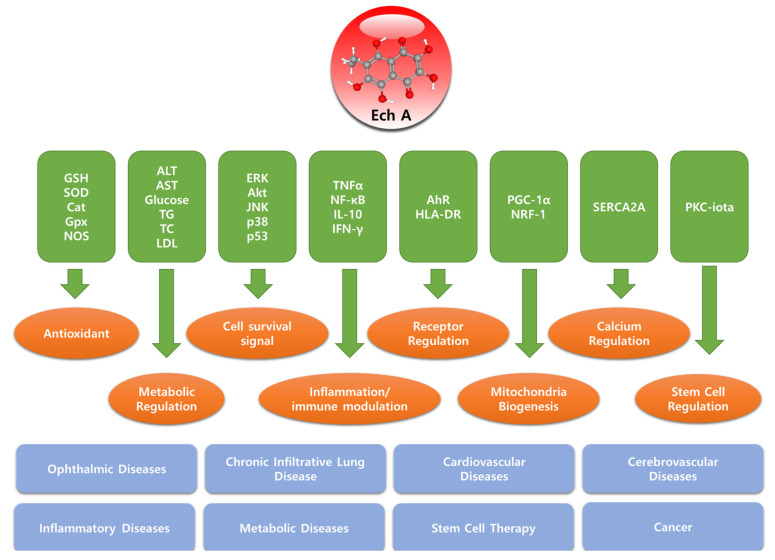
Molecular targets, biological functions, and target diseases of Echinochrome A.

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
