# Peer review of "Multifaceted Clinical Effects of Echinochrome"

_marinedrugs, 2021, doi:10.3390/md19080412_

Round 1

Reviewer 1 Report

Overall, this is a well-written and thoughtful review of the effects of echinochrome.  There are an incredible number of therapeutic effects and potential applications for this marine pigment.  Below are some points to consider prior to publication.

  1. Between lines 84-113, there are no specific citations of experimental work. Are all of these findings attributed to citation 21?
  2. In most cases throughout the manuscript, it is not clear whether echinochrome (Histochrome) treatment is compared to placebo. Are the effects due to echinochrome, or simply to normal healing?  I would assume that these studies used some form of control group?  Please clarify.
  3. Lines 230-234: Is this referring to the control group in the cited #29 study? If so, it should not be set aside as a separate paragraph.
  4. Lastly, some global perspective / suggestion on why echinochrome has such wide-ranging benefits to human health would be appreciated. We know several details from the studies, but do the authors have any thoughts, or are there any accepted theories for why these effects appear to be so strong?  Could there be a basis in the evolution of marine species vs. land-based evolution?

Author Response

  1. Between lines 84-113, there are no specific citations of experimental work. Are all of these findings attributed to citation 21?

Answer: Everything between lines 84-113 is the large-scale clinical study published in the reference [21].

We added to text:

“Thus, the authors of this large-scale clinical study conclude that the drug was effective in 92.8% of the cases with proliferative processes, degeneration, and hemophthalms [21].”

  1. In most cases throughout the manuscript, it is not clear whether echinochrome (Histochrome) treatment is compared to placebo. Are the effects due to echinochrome, or simply to normal healing?  I would assume that these studies used some form of control group?  Please clarify.

Answer: Thank you very much for the critical comment. In most of the studies presented in this review histochrome treatment was compared with standard therapy and with other antioxidant drugs (in some studies). All the effects described are due to histochrome treatment. We added this issues in the discussion part as below

In the past several decades, histochrome, a drug formulation of Ech A, has shown limited clinical use in Russia despite its various pharmacological effects. In most of the studies presented in this review histochrome treatment was compared with standard therapy and with other antioxidant drugs (in some studies) and has proven its advantage. The study results on the various pharmacological actions of Ech A and its derivatives provide important theoretical evidence for expanding the clinical application of histochrome

  1. Lines 230-234: Is this referring to the control group in the cited #29 study? If so, it should not be set aside as a separate paragraph.

Answer: This paragraph was a part of introduction to the chapter 2.2.2, but we think it would be better to delete, since this thought has already been reflected above.

  1. Lastly, some global perspective / suggestion on why echinochrome has such wide-ranging benefits to human health would be appreciated. We know several details from the studies, but do the authors have any thoughts, or are there any accepted theories for why these effects appear to be so strong?  Could there be a basis in the evolution of marine species vs. land-based evolution?

Answer: Thank you for the valuable comment. We added some our discussion points to Conclusion sections:

Even though Ech A looks like a simple molecule it is not so simple. First of all, it is highly oxygenated, oxygen makes up 42% of the mass of the molecule in contrast to most other secondary metabolites especially of terrestrial origin. Second, Ech A has the ethyl side chain what makes the molecule lipophilic and able to enter membranous structures. Also due to these structural features Ech A become ionisable under physiological conditions and forms 8 possible protomeric structures [new reference 82]. Most likely that such a variety of strong effects is explained exactly by the unicity of the structure of Ech A.

Reviewer 2 Report

The authors reviewed the effects of echinochrome on  many diseases in the clinical setting. They summarized a huge data nicely and this review seems to be educative.

Major problem: None.

Minor problems:

1.  In page 8, line 323: The authors adopted "ischemic heart disease" as the subtitle in the section, however, the authors described "coronary heart disease" here. If the authors unify these words, the readers will understand it easily.                                                                                                                                                                                                                                                               2. In page 9, line 395-397. This sentence seems to be unclear. It is clear that doxorubicin has cardiotoxicity, however, sodium nitroprusside, which is an anti-hypertensive drug, seems to be uncertain whether this drug has cardiotoxicity. The authors should comment on this. Furthermore, if they insist so, they should put some references here.

Author Response

  1. In page 8, line 323: The authors adopted "ischemic heart disease" as the subtitle in the section, however, the authors described "coronary heart disease" here. If the authors unify these words, the readers will understand it easily.

Answer: Thank you very much for the suggestion. The study pointed out by the reviewer is the result of confirming the cardiac ATP conservation effect of Ech A pretreatment in isolated myocardial tissue from patients with coronary artery disease and exposing it to an ischemic environment secondarily. Therefore, specifying the disease as coronary heart disease will explain the contents of this paper well.

  1. In page 9, line 395-397. This sentence seems to be unclear. It is clear that doxorubicin has cardiotoxicity, however, sodium nitroprusside, which is an anti-hypertensive drug, seems to be uncertain whether this drug has cardiotoxicity. The authors should comment on this. Furthermore, if they insist so, they should put some references here.

Answer: Thank you for the valuable suggestion. It was suggested that sodium nitroprusside (SNP) is used clinically as a rapid-acting vasodilator and in experimental models as donor of nitric oxide (NO). High concentrations of NO have been reported to induce cardiotoxic effects including apoptosis by the formation of reactive oxygen species (Chiusa et al. Eur J Histochem. 2012). Although sodium nitroprusside (SNP) is an effective hypotensive drug and is often used in pediatric intensive care units and to treat acute heart failure, clinical application of SNP is limited by its cardiotoxicity (Lee et al. Cell Biol Int. 2014). I added related references there.

Several therapeutic drugs, such as sodium nitroprusside (SNP), a hypertensive drug, and doxorubicin, an anticancer drug, can induce oxidative stress and cardiotoxicity, which are adverse risk factors for heart failure [53, 54].

  1. Chiusa, M.; Timolati, F.; Perriard, J. C.; Suter, T. M.; Zuppinger, C., Sodium nitroprusside induces cell death and cytoskeleton degradation in adult rat cardiomyocytes in vitro: implications for anthracycline-induced cardiotoxicity. Eur J Histochem 2012, 56, (2), e15.
  2. Lee, S. R.; Lee, S. J.; Kim, S. H.; Ko, K. S.; Rhee, B. D.; Xu, Z.; Kim, N.; Han, J., NecroX-5 suppresses sodium nitroprusside-induced cardiac cell death through inhibition of JNK and caspase-3 activation. Cell Biol Int 2014, 38, (6), 702-7.
